# Photobiomodulation for Lowering Pain after Tonsillectomy: Low Efficacy and a Possible Unexpected Adverse Effect

**DOI:** 10.3390/life12020202

**Published:** 2022-01-28

**Authors:** Fulvio Celsi, Paola Staffa, Martino Lamba, Veronica Castro, Maddalena Chermetz, Eva Orzan, Raffaella Sagredini, Egidio Barbi

**Affiliations:** 1Institute for Maternal and Child Health-IRCCS “Burlo Garofolo”, 65/1 Via dell’Istria, 34137 Trieste, Italy; paola.staffa@burlo.trieste.it (P.S.); martino.lamba95@gmail.com (M.L.); veronica.castro@burlo.trieste.it (V.C.); maddalena.chemetz@burlo.trieste.it (M.C.); eva.orzan@burlo.trieste.it (E.O.); raffaella.sagredini@burlo.trieste.it (R.S.); egidio.barbi@burlo.trieste.it (E.B.); 2Department of Medical, Surgery and Health Sciences, University of Trieste, 34149 Trieste, Italy

**Keywords:** adenotonsillectomy, clinical trial, nociception, low-level laser therapy, tonsils

## Abstract

Background: Tonsillectomy is one of the most common surgical procedures performed in children as a treatment for obstructive sleep apnea due to tonsil hypertrophy or highly recurrent tonsillitis. Odynophagia, associated with food refusal for the first few days, is a common post-operative complaint. Available drugs for pain management, while efficacious, present some drawbacks, and a novel strategy would be welcome. Photobiomodulation (PBMT), in this context, can represent a possible choice, together with pharmacological therapy. The aim of this study was to evaluate PBMT effects compared to standard pain therapy on nociceptive sensation at different time points and administration of painkiller. Methods: A registered, controlled, randomized, double-blind clinical trial was performed. Twenty-two patients were recruited and divided into laser-treated (T) or untreated (UT) groups, based on random assignment. In T group, immediately after tonsillectomy, performed with cold dissection technique, laser light was applied to the surgery site (using a Cube 4 from Eltech K-Laser s.r.l., Treviso, Italy), and then hemostasis was performed using bismuth subgallate paste. In C group, the same procedure was performed, except that laser light was switched off. The primary outcome was the difference in pain scores between subject receiving photobiomodulation (PBMT) and subjects receiving standard care after 24 h; the secondary outcomes were pain scores at awakening and at 48 h together with distress (delirium) at awakening. Results: Two patients from the T group experienced a post-surgery bleeding, and one of them required revision of the hemostasis under general anesthesia. A preliminary analysis of pain sensation reported by the patients or caregivers did not show differences between treated and untreated subjects. Conclusion: These data suggest that PBMT could increase post-surgical bleeding.

## 1. Introduction

Tonsillectomy is one of the most common surgical procedures performed for children under 15 years of age [1], as a treatment for recurrent tonsillitis or respiratory problems due to tonsil hypertrophy. Odynophagia is common after surgery resulting in partial or complete refuse of food for the first days and decreased activities. Children usually recover in a variable period from 4 to 7 days, but in some cases, odynophagia can be severe enough to limit oral fluid intake up to dehydration requiring intravenous fluids administration [2]. Current protocols for pain management allow a good pain control but still there is some debate about the risk of bleeding related to the use of NSAIDS, and of respiratory depression related to the use of opioids.

Multiple strategies in pain management could be considered; in this context, photobiomodulation (PBM) represents a possible choice. In different clinical context, PBM has been demonstrated to improve tissue regeneration [3], to lower inflammation, and to diminish pain sensation. In some cases, PBM is the first therapeutic choice, specifically in chemotherapy-induced oral mucositis: a systematic review [4] indicated that prophylactic PBM lowered inflammation and pain. Furthermore, PBM has been demonstrated to be effective in management of post-operative pain [5], and a pilot study showed that, after thoracic surgery, PBM can significantly lower pain sensation [6].

PBM mechanisms of action at cellular level are different and not completely elucidated. Mitochondria appear to be the primary site of response to laser illumination, that acts on cytochrome c oxidase (CCO). Nitric oxide (NO) production can be associated with this enzyme and PBM (specifically in near infrared wavelengths) liberates it, increasing CCO activity and thus reactive oxygen species (ROS) production. Increased ROS levels induce cellular metabolism, with different responses in the diverse cellular types [7]. Using a sensory neuronal cell model, a recent study showed that near infrared (NIR) PBM can increase ROS level and diminish physiological response to capsaicin, a natural inducer of painful stimulus [8]. Thus, in this cellular model, ROS increase induced by NIR PBM could be linked to reduced nociceptive signaling.

A previous study evaluated PBM efficacy in managing pain sensation after pediatric tonsillectomy and it demonstrated that in 9 patients treated with laser; nociception was significantly lower requiring fewer analgesic drugs in respect to untreated children (laser wavelength 685 nm, spot size 2 mm^2^, and fluence 4 J/cm^2^) [9]. A more recent study [10] demonstrated, in 60 adults undergoing tonsillectomy, a significant reduction in pain sensation and in administration of analgesic drugs (laser wavelength 980 nm, spot size 2 mm^2^, and fluence 4 J/cm^2^). The use of this treatment, which is non-damaging, non-toxic, and easy to supply, could improve individual quality of life after a surgical treatment. Therefore, this study aimed to evaluate if pain sensation after tonsillectomy differs between patients treated with PBM or not, while standard analgesic practice was still administered to both groups.

## 2. Materials and Methods

### 2.1. Study Design

A randomized, double-blind, controlled study was designed (trial registration number: NCT04693208). Patients were divided in two experimental arms, treated with PBM (T) or untreated (UT), based on random assignment made using online software (http://www.randomization.com, accessed on 5 November 2019). To obtain a balanced number of subjects assigned either to T or UT groups, a block randomization was used. Twenty-two patients were enrolled before stopping the trial.

### 2.2. Ethics

The study was submitted and approved by the regional ethic committee (CEUR-2020-Sper-065). All parents or caregivers of patients eligible for the study were informed by study assessors of the objective and were required to complete the consent form.

### 2.3. Eligible Criteria

Inclusion criteria:Age >3 and <18 years;Tonsillectomy or adenotonsillectomy due to apneic roncopathy, and/or obstructive sleep apnea, and/or recurrent tonsillitis;Exclusion criteria:ASA Score > 2;Neuropsychiatric comorbidity;Pro-hemorrhage coagulation disorders.

### 2.4. Population Characteristics

Population demography is described in Table 1.

### 2.5. Procedures

During the routine pre-operation meeting, performed with the patient’s parent or caregiver, possibility of participation to the clinical trial was offered. If the parent or caregiver agreed, the day of the operation, a researcher—not involved in recruiting the patient—opened a sequentially numbered, sealed, and opaque envelope, based on the sequence obtained by the software. Inside the envelope was the allocation to UT or T groups and the researcher together with the surgeon performed PBM or placebo treatment. The surgeon was unaware of the random assignment result, and laser light was switched on or off by the researcher.

### 2.6. PBM Protocol

PBM was executed using a Cube 4 device from Eltech K-Laser s.r.l., (Italy) in pulsated mode, with parameters indicated in Table 2 (recommended by the producer); the tissue was exposed to each wavelength sequentially, without pauses. The laser tip diameter corresponds to the spot size (2 cm^2^), and it was constantly moved over the surgical area at an approximate distance of 1 cm by the operator to avoid thermal effect, during the total 3 min of laser illumination in a single session (Figure 1 indicates the approximate area of illumination).

Placebo treatment was performed by passing the laser probe in the surgical area with light switched off and only light guide on (to simulate laser emission) for the same amount of time. Laser switching on or off was performed by a researcher aware of the allocation group, while the surgeon remained oblivious to the treatment performed.

After either PBM or placebo treatment, hemostasis was performed using bismuth subgallate paste. During surgery, Fentanyl 2 μg/kg and Perfalgan 10 mg/kg (if patient weight <10 kg) or 15 mg/kg (if patient weight >10 kg) were administered. After hemostasis, anesthesia was interrupted and patients were moved to an observational room, were the Pediatric Anesthesia Emergence Delirium (PAED) scale and Face–Legs–Activity–Cry–Consolability (FLACC) scales were evaluated upon patients awakening by a trained nurse, unaware of the treatment group. After 24 h and 48 h, FLACC scale was further evaluated by a trained researcher, also unaware of the treatment group, to assess change in pain perception.

## 3. Results

During the clinical trial, out of ten patients, two experienced post-surgical bleeding. The 1st had a minor bleeding 48 h after the procedure and was admitted for 1 night monitoring, requiring no specific treatment. The 2nd patient had minor bleeding at 12 h, and a subsequent further bleeding occurring during prolonged postoperative monitoring, 36 h after surgery. The second bleeding required revision of the hemostasis under general anesthesia. The intraoperative findings showed a tendency to bleeding in several points of the surgery area from both sides, and a definitive hemostasis was obtained by bipolar cauterization and bismuth subgallate. Both children tested negative for possible bleeding defects (Table 3). The patients’ history was negative for previous bleeding episodes or easy bruising; they had not received NSAIDS treatment.

In the control group, no post-surgical bleeding was observed. The rate of post-surgical bleeding requiring any increase in level of care such as prolonged hospital admission, ED readmission for observation, surgical re-evaluation, or need for a specific treatment for tonsillectomy was 0 out of 715 patients between 2019 and 2020 for tonsillectomy. 

PAED values at awakening indicates no difference between groups (Wilcoxon test, *p* = 0.12) (Figure 2a). FLACC values (relative to pain sensation) at awakening, after 24 h, and after 48 h showed no significant difference (Wilcoxon test, *p* = 0.19 at awakening, *p* = 0.14 at 24 h, and *p* = 0.56 after 48 h; Figure 2b–d, respectively). Due to the small number of patients, power for PAED evaluation at awakening is 0.74, while for FLACC it was 0.6 at awakening and 0.58 and 0.22 for 24 h and 48 h evaluation time, respectively.

## 4. Discussion

This prospective trial was stopped due to two episodes of bleeding in the first ten patients of the laser treatment group, which suggested a possible causal relationship between treatment and the adverse event. 

This decision was taken since no bleeding occurred in the control group, as well as the fact that bleeding is an extremely rare occurrence in our series. While the literature reports a rate of primary hemorrhage within 24 h from 0.2% to 2.2% and secondary hemorrhage (after 24 h) varies from 0.1% to 3 [11], in our institute, a total of 715 patients were admitted between 2019 and 2020 for tonsillectomy and none of them experienced post-surgical bleeding requiring any increase in level of care, such as prolonged hospital admission, ED readmission for observation, surgical re-evaluation, or need for a specific treatment. Remarkably, no patient in our trial was treated with NSAIDS, and a possible coagulation defect was ruled out in both bleeding patients.

A possible mechanistic explanation between PBM and bleeding in this setting points to the key role of nitric oxide (NO). This molecule has different biological roles and was identified as an endothelium-derived relaxing factor [12] regulating vascular tone and blood flow by activating soluble guanylate cyclase (sGC) [13]. Moreover, NO inhibits cytochrome c oxidase, a key enzyme in mitochondrial respiration, lowering ATP production [14], and it was supposed that laser light could photo-dissociate NO from its binding site on the enzyme and thus release it in the cellular milieux [15]. Thus, laser light can increase mitochondrial respiration rate and increase NO concentration. Moreover, in isolated dorsal root ganglia neurons (DRGs), PBM can stimulate both NO release and increase NO synthase expression [16], a mechanism hypothesized to be linked to analgesia induced by laser stimulation. It is thus possible that NO released by PBM induces vascular relaxation and increase in microcirculation; a recent report indeed demonstrated that PBM (specifically near-infrared wavelength) can increase palm microcirculatory flow by arteriolar vasodilatation in some subjects [17], suggesting that a similar phenomenon may happen on the surgical site in tonsillectomy. In this specific hypothesis, the role of bismuth subgallate in this series, which has been shown to suppress production of NO and PGE2 in a dose-dependent manner, should be considered as a protective one against bleeding.

As far as the impact of PBM on pain is concerned, the early interruption of the trial does not allow a conclusion. In vitro findings suggested that PBM can reduce calcium influx after capsaicin stimulation [18], blocks fast axonal flow and mitochondrial membrane potential [19], and lowers bradykinin-induced response in cultured neurons [20]. PBM has also been shown to have analgesic effect in different clinical context, such as carpal tunnel syndrome [21] or temporomandibular disorder [22]. A meta-analysis conducted in 2019 indicates that PBM can be effective in modulating pain and inflammation following surgery, with the most efficacious results for postoperative pain after tonsillectomy [23]. Additionally, considering two previous studies that indicated a beneficial effect on pain modulation in tonsillectomy in adult [10] and pediatric populations [9], this clinical trial was conceived and proposed to the ethical committee. The PBM protocol proposed in this study differs from both studies cited above, in using 3 different wavelengths instead of 1, either 980 nm for adults or 685 nm for children, and a higher fluency (12 J/cm^2^ in our protocol compared with 4 J/cm^2^). However, this protocol is routinely used to increase recover of surgical wounds, especially after tooth extraction, and it was chosen for these reasons. Remarkably, no important adverse effects were reported by the literature, considering also that PBM was already used in pediatric patients with “difficult”, open lesions, such as oral mucositis [24], using a single wavelength, 635 nm, with 4J/cm^2^ fluence.

A preliminary analysis of pain perception scales suggests a trend toward lower levels of nociception, although not reaching statistical significance, this was in agreement with previous studies on tonsillectomy and other kinds of chronic pain. We also performed a linear regression analysis without reaching statistical significance (data not shown), Power analysis indicated that the scarce number of patients limits the possibility that the results are conclusive, and it suggest the need of more data, but, unfortunately, enrolment of more patients was halted due to the possible side effects of laser therapy.

## 5. Conclusions

This report suggests a possible correlation between PBM (using three different wavelengths) and post-tonsillectomy bleeding. While PBM can be a useful intervention in reducing chronical or post-surgery pain, as various studies have been demonstrated, using these specific wavelengths settings may be responsible for post-surgical bleeding. As previously shown, PBM could be useful for reducing post-tonsillectomy pain, although using different wavelengths. Usefulness of PBM as a pain-reducing therapy remains undisputed, while the present work highlights a possible unexpected side effect of the specific PBM protocol used. Further studies are needed to better clarify the link between this PBM protocol and increased bleeding by possibly acting on blood flow and microcirculation.

## Figures and Tables

**Figure 1 life-12-00202-f001:**
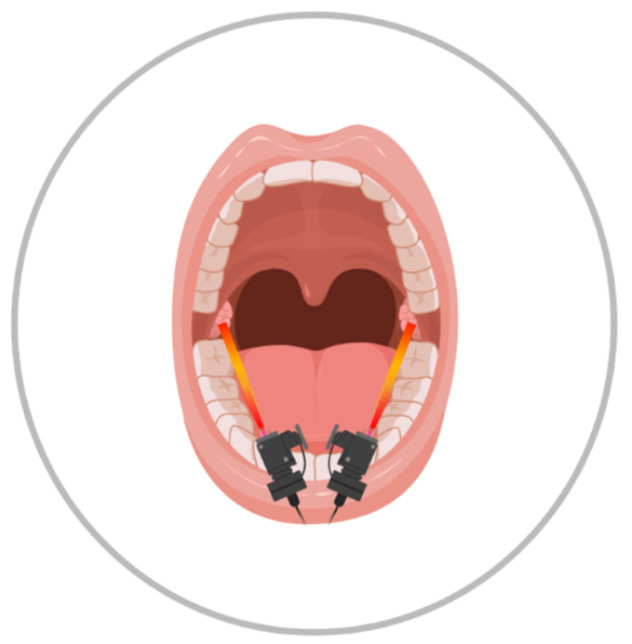
Schematic drawing of laser’s site intervention.

**Figure 2 life-12-00202-f002:**
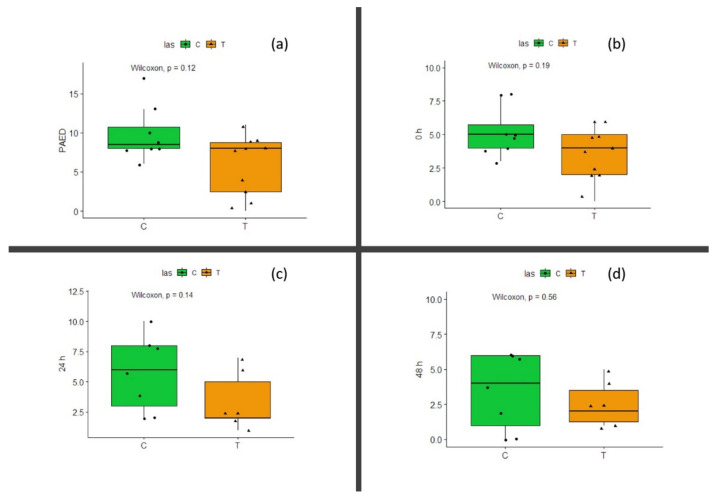
(**a**) PAED scale evaluated at patients awakening: UT—untreated patients; T—laser-treated patients. (**b**) FLACC scale evaluated at patients awakening: UT—untreated patients; T—laser-treated patients. (**c**) FLACC scale evaluated 24 h after surgery: UT—untreated patients; T—laser-treated patients. (**d**) FLACC scale evaluated 48 h after surgery: UT—untreated patients; T—laser-treated patients. Wilcoxon test results are shown above the graph; *p* is not significant for all the considered evaluation.

**Table 1 life-12-00202-t001:** Population characteristics.

	Median Age	Min Age	Max Age	IQR	nr
UT	5	3	16	2	10
T	5	3	8	2	12

UT—untreated patients; T—treated (with PBM) patients; Median, Min, and Max Age: Median, Minimum, and Maximum patient age, respectively; IQR—interquartile range; nr—number of patients in each group.

**Table 2 life-12-00202-t002:** PBM protocol parameters.

Wavelength	Mean Power	Peak Power	Fluence	Frequency	Duration
445 nm	0.1 W	0.2 W	6 J/cm^2^	4 Hz	60 s
660 nm	0.1 W	0.2 W	6 J/cm^2^	4 Hz	60 s
970 nm	0.2 W	0.4 W	12 J/cm^2^	4 Hz	60 s

**Table 3 life-12-00202-t003:** Laboratory results for coagulation tests. All the parameters are within normal range.

*Test*	Patients	Values	Ref. Value
*Platelets count*	1	253 × 10^3^ /mL	150–400
	2	422 × 10^3^ /mL	150–450
*Prothrombin time (INR)*	1	1.13	0.80–1.20
	2	1.15	
*Prothrombin time (Sec)*	1	NA	
	2	13.8 s	
*Prothrombin time (ratio)*	1	1.13	0.85–1.15
	2	1.15	0.78–1.20
*Activated partial tomboplastin time (APTT) (sec)*	1	NA	
	2	30.8 s	
*Activated partial tomboplastin time (APTT) (ratio)*	1	1.13	0.85–1.15
	2	0.96	0.76–1.18
*Fibrionogen (Clauss)*	1	404 mg/dL	180–380
	2	346 mg/dL	160–380
*Antithrombin (chromogenic)*	1	NA	
	2	91%	78–124%
*Coagulation Factor XIII*	1	106%	70–130%
	2	NA	NA
*Von Willebrand antigen*	1	128%	Group 0 = 40–140% Group A, B, AB = 65–165%
	2	NA	NA
*Von Willebrand activity*	1	97%	Group 0 = 40–130% Group A, B, AB = 50–160%
	2	NA	NA

## Data Availability

The data presented in this study are available on request from the corresponding author. The data are not publicly available due to privacy restriction.

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
