# Peer review of "Photobiomodulation for Lowering Pain after Tonsillectomy: Low Efficacy and a Possible Unexpected Adverse Effect"

_life, 2022, doi:10.3390/life12020202_

Round 1
Reviewer 1 Report
Good and interesting article, even that the number of cases is low.
Please check Tabel 1 and Figure 2,
Is not clear from the methodology if the protocol used for PBM is recommended also by the producer of the laser or not, can you please be more clear about that.
Reviewer 2 Report
The manuscript entitled " Post-tonsillectomy bleeding after photobiomodulation: an un-expected adverse effect " deals with the evaluation PBMT effects compared to standard pain therapy on nociceptive sensation at different time points and administration of painkiller.
The study is very interesting; however, major changes need to be addressed.
The title of the study did not cover obtained results. In two cases of 22, the authors found bleeding after tonsillectomy and claim that PBM evokes this complication. This conclusion is unclear and should be corrected.
Keywords: should be in alphabetical order, KEYWORDS should not contain the same words that are within the title of the text. Thus these should be changed appropriately
Abstract: Add details of the manufacture name of the laser as well as all the parameters (power, energy dose per point, total dose frequency, pulse duration, tip diameter, and distance to the target, etc…)
Introduction
Line 46-48, Add a citation to this statements “In different clinical context PBM has been demonstrated to improve tissue regeneration, to lower inflammation and to diminish pain sensation..”
doi.org/10.1089/photob.2020.4863 doi.org/10.3390/ma13102265
Line 62 Remove a dot at the end of the sentence.
M&M
The laser parameters were not described well.
Add the laser’s manufacturer and all important parameters e.g. power, spot size, mean power density (irradiance), wave mode, energy per point, radiant exposure:, time per point, the total energy dose (radiant energy) per session, tip diameter, when the PBM was applied, how many sessions? What was cumulative radiant energy?
The Conclusion should be based on p< 0.05 results, not based on the two case reports.
Reviewer 3 Report
1- Extensive editing of English language and style required.
Example: "these author equally contributed equally to this paper".
2- In the abstract
"It is apnea and not apnoea."
"...strategy will be welcomed and not welcome"
"The aim of this study" and not "aim of this study".
3- In the Introduction
The introduction must be improved by a better explanation of the:
- Mechanism of action of Photobiomodulation therapy (acts on the cytochrome c oxidase, ROS...).
- Mechanism of action in which PBM affects the pain.
- What is the null hypothesis of this study ?
4- In the material and methods
- It is not explained clearly how was possible to achieve the double blind randomization ? The operator who performed the PBM was not aware if the laser was used or not ? Please explain more deeply.
- In the study design section, add the number of patients ? It was hard for me to know the number of patients and I had to refer to the abstract section.
- The PBM parameters are not described in details. Today it is not enough to only mention the power of the laser and the energy density (fluency). the tip diameter, the delivery mode (contact non-contact) and (continuous and non-continuous) must be added (please add all the details about the laser parameters).
- A table showing the irradiation parameters will make it easier for the reader to read.
- Both the 445 nm and 970 nm are not indicated today for the Photobiomodulation therapy. (Today it is recommended to use wavelengths around and between 600 and 690 nm).
- There is no Statistical Analysis section. How was the comparison made ?
In its current form the manuscript does not meet the standards of a scientific paper.
Round 2
Reviewer 2 Report
Dear authors, thank you for the revision of the paper,
In Table 2, there are some mistakes in units of the peak and mean energy; the correct unit is Joul.
Reading the methodology section is difficult to understand how various types of laser wavelengths (three lasers) were applied.
Why were so many wavelengths with different characteristics of interaction with tissues used?
Does the post-surgical bleeding was noticed for only one wavelength?
The use of so many wavelengths with different penetration levels and differences in tissue absorption can also affect the pain level.
This should be discussed.
Author Response
"Please see the attachment."
